# Theoretical, Numerical and Experimental Assessment of Temperature Response in Polylactic Acid and Acrylonitrile Butadiene Styrene Used in Additive Manufacturing

**DOI:** 10.3390/polym14091714

**Published:** 2022-04-22

**Authors:** Camen Ema Panaite, Andrei-Marius Mihalache, Oana Dodun, Laurențiu Slătineanu, Aristotel Popescu, Adelina Hrițuc, Gheorghe Nagîț

**Affiliations:** 1Department of Mechanical Engineering and Road Automotive Engineering, “Gheorghe Asachi” Technical University of Iași, Blvd. D. Mangeron, 43, 700050 Iași, Romania; carmen-ema.panaite@academic.tuiasi.ro (C.E.P.); aristotel.popescu@academic.tuiasi.ro (A.P.); nagit@tcm.tuiasi.ro (G.N.); 2Department of Machine Manufacturing Technology, ”Gheorghe Asachi” Technical University of Iaşi, Blvd. D. Mangeron, 59A, 700050 Iasi, Romania; andrei.mihalache@tuiasi.ro (A.-M.M.); oanadodun@tcm.tuiasi.ro (O.D.); slati@tcm.tuiasi.ro (L.S.)

**Keywords:** heat transfer, polymer materials, 3D printing, influencing factors, empirical mathematical model, experimental Taguchi L8 program

## Abstract

A better understanding of heat transfer through materials used for 3D-printed parts could lead to an extension and an optimization of their use. A topic of interest could be analyzing temperature variation in these materials during cooling processes. Experimental research and equipment were designed to obtain additional information on the surface temperature decrease when the opposite wall surface is exposed to a freezing temperature. Experimental tests were performed on samples made of polylactic acid (PLA) and acrylonitrile butadiene styrene (ABS). An experimental Taguchi L8 program was used, with seven independent variables at two levels of variation. The experimental data analysis with specialized software based on the least-squares method identified a mathematical model of first-degree polynomial type. The coefficients for each input factor involved provide information on the magnitude and trend of the considered output parameter when the input factors’ values change. It was found that the thickness of the 3D printing layer, the thickness of the test sample, and the 3D printing speed are the main factors that affect the temperature decrease rate.

## 1. Introduction

Thermal conductivity is a physical property that evaluates the ability of a material to conduct heat. Its importance becomes significant when choosing proper materials for different applications. For cooling or heating purposes, materials with higher thermal conductivity are chosen, while lower thermal conductivity materials are recommended for insulation applications.

Various methods are employed to determine the thermal conductivity of materials, mainly based on measuring temperatures on opposite walls of a test sample of known material and dimensions under a specified heat flux. Methods used to evaluate the thermal conductivity of materials include the guarded hot plate method, axial flow method, cylinder method, flash method, hot wire method, needle probe method, and transient plane method [1,2,3,4,5,6,7,8]. 

Metallic materials are good heat conductors, as temperature variation on one surface causes a relatively rapid temperature variation on its other surfaces. On the contrary, materials that are poor heat conductors are considered thermally insulating. This category includes polymeric materials, usually with low thermal conductivity values.

The last decades have highlighted an expansion of polymeric material usage in many areas (automotive or other means of transportation [9,10,11], household items [10,12,13,14], industrial equipment [15,16], medicine [17,18,19], etc.). Moreover, there is a clear trend toward expanding the fabrication of polymer parts through 3D printing processes. As 3Dprinting equipment patents expired more than two decades ago, many companies got involved in the industrial manufacturing of 3D printers.

3D printing is part of the broader category of additive manufacturing processes, i.e., parts are obtained gradually by successive addition of new layers of a material. 3D printing processes developed further when software advancements made part dimensional characterization possible with adequate accuracy, which is crucial for successive layer deposition.

One of the 3D printing processes is based on melting a polymer wire and gradual molten material deposition, with the part being generated from successive layers. The shape of each layer is generated by complex relative movement between the nozzle with the molten material and the printer base plate. This is called the fused filament fabrication method.

The analysis of some practical aspects of this manufacturing method revealed several input factors that influence the material properties of the final product. Therefore, researchers performed more in-depth investigations on fused filament additive manufacturing processes regarding input factors that influence different physical and mechanical properties of the 3D-printed part, such as thermal conductivity. 

There is a special interest in using polymeric materials for thermal insulation of residential, commercial, or industrial buildings to reduce the effects of excessive cold or heat. Such materials’ low heat transfer conductive capacity may represent a significant argument for using polymers as thermal insulating materials [8,9,15,16,20,21,22,23,24,25,26]. However, the material properties of a part manufactured by 3D printing can be influenced by thermal interactions during the 3D printing process between the process components (filaments, printing support, environment) [20].

Elkholy and Kempers evaluated the thermal properties of polymers test samples manufactured by 3D printing [27]. The test sample is placed between two blocks, one for cooling and one for heating, and the blocks’ temperatures are determined. Elkholy et al. developed research that aimed to highlight the anisotropic behavior of some samples made by fused filament fabrication [28].

In a review paper on expanded polystyrene in building construction, Ramli Sulong et al. also highlighted this material’s good thermal insulation capacity [21].

Extensive research completed by a doctoral thesis that addresses issues related to the thermal insulation capacity of 3D-printed materials was conducted by M. Harris [22]. Experimental research was performed on samples of polylactic acid and blends of high-density polyethylene and polypropylene with different thermoplastics (acrylonitrile butadiene styrene and polylactic acid).

Doğan and Tan [23] theoretically investigated (using a model based on finite elements and ANSYS software) and performed experimental research on the thermal behavior of some expanded polystyrene blocks.

An evaluation of the thermal insulation capability of some polystyrene foam structures containing rice hulls was made by Flores [29]. The author estimated that the density of a material could significantly influence the thermal insulation value. Polystyrene is currently used as a thermally insulating material where appropriate conditions are available and there are no health risks.

Eom et al. conducted experimental research to highlight the behavior of the thermal insulation capacity of structures made of 3D-printed thermoplastic polyurethane [24]. They found that increasing the air gap is more effective than increasing the pore size.

Grabowska and Kasperski addressed the issue of structure optimization for 3D-printed polypropylene multilayer films, with closures of different shapes and sizes [25]. The authors found that the best thermal insulation corresponds to single-layer quadrangle and hexagonal closures, with a structural density of 180 kg/m^3^. The thermal conductivity was 0.0591 W/(m∙K).

T. de Rubeis addressed the issue of manufacturing 3D-printed blocks used as thermal insulation materials for buildings [26]. The author used a specially designed hot box to assess the thermal insulation capability to ensure known, repeatable, and steady thermal experimental conditions. The infrared thermography technique and heat flow meter method were also used to characterize the thermal insulation of the 3D-printed blocks.

Other published research assessed the ability of polymeric materials as a heat transfer medium or issues involving the thermal conductivity of polymers [30,31].

Given the fairly widespread use of polystyrene as thermal insulation for buildings, researchers have been concerned with better characterizing the behavior of this material from the point of view of thermal conductivity [21,29,32,33,34,35].

The research presented in this paper aimed to highlight the influence of some factors regarding test sample thickness or the influence of manufacturing conditions on a temperature decrease for test samples of polylactic acid (PLA) and acrylonitrile butadiene styrene (ABS), respectively. It also identified empirical mathematical models that would provide information on the temperature decrease rate for changing values of several factors defining test sample thickness and the conditions of the 3D printing process. 

## 2. Materials and Methods

### Assumptions Regarding Heat Transfer in Plastic Parts Made by Additive Manufacturing

The quality of thermally insulating materials of 3D-printed parts can effectively refer to the decrease in heat transfer rate for different materials.

The test arrangement shown in Figure 1 was considered. A flexible-wall package containing a low-temperature liquid solution (gel coolant) was used. The package flexibility enables contact over a large area between the coolant and the test sample surface of the 3D-printed material.

This research aimed to develop diagrams of temperature variation on the free surface of the 3D-printed test samples manufactured with different adjustments to the printer’s working parameters. Measurements were performed on the test sample’s free flat surface opposite the cooled surface. The flexible-wall package ensured that the gel coolant was initially at −15 °C. It is expected that the temperature of the free flat surface will gradually decrease due to heat transfer through the test sample material. The rate of this process depends on the material thermal properties, especially its thermal conductivity.

There are heat gains towards the gel-coolant package during measurements from the test sample and from the environment. All components shown in Figure 1 were enclosed in a wooden box (low thermal conductivity) and insulated with polystyrene foam to minimize the latter ones. To ease the process of changing the test samples and cooling packages, the box was provided with two doors with holes in the central areas; these doors were also used to perform periodic temperature measurements using IR thermometers (Figure 2).

The test materials were PLA and ABS. They are two polymeric materials frequently used in 3D printing processes.

The PLA filament used in this experimental research was brown and was delivered by Fillamentum Manufacturing Czech s.r.o. (Hulín, Czech Republic). The filament of the second material (white ABS) was produced by the manufacturer of the 3D printing equipment (Ultimaker, Utrecht, Netherlands). The use of ABS filament made by the manufacturer of the 3D printing equipment was preferred, considering that this is a safer solution and verified by the manufacturer. It was estimated that more rigorous requirements would accomplish ABS test sample manufacturing by printing.

Observation. The glass transition temperature is when the material changes from a brittle, hard state to a soft rubbery state. The Differential Scanning Calorimetry (DSC) evaluates the heat flow and compares the amount of heat supplied to the test sample with a similarly heated “reference” to establish the transition points. The Dynamic Mechanical Analysis (DMA) evaluates a material’s response to an applied oscillating stress and the influence of temperature and frequency on that response [36].

ABS white is a mixture of acrylonitrile-co-butadiene-co-styrene, polyethylene terephthalate, and polycarbonate. The manufacturers indicated the characteristics of the two materials. Thus, PLA is an industrial composter containing more than 98% polylactide resin. Similar information and compositions on the two filament materials (ABS and PLA) are also provided by other manufacturers of materials used in 3D printing. Some physical properties of the two materials analyzed in this research are presented in Table 1.

Theoretically, a model of heat transfer following the first law of thermodynamics [39] yields the equation:(1)$$\frac{dE_{\Omega }}{dt}=P_{str}+Q_{exch},$$

A mathematical model such as Equation (1) can be used to detect possible thermal degradation of polymeric material when chemical changes occur due to exposure to higher temperatures in the absence of oxygen, where *E*_Ω_ is the internal energy, *t* is the time, *P_str_* is the stress power, and *Q_exch_* is the exchanged heat rate. This theoretical relationship considers the stress power factor when the power is converted into heat by dissipation [39].

## 3. Results

### 3.1. Finite Element Modeling of Thermal Conductivity Variation

The finite element analysis models the temperature variation within the test sample for a certain time after the flexible-wall package with gel coolant was placed on the sample side surface. Using the ANSYS R19.2 software, the graphical representation in Figure 3 provides an image of temperature distribution in a PLA test sample.

Test samples of various thicknesses were subjected to a gel coolant at a temperature of about −15 °C on one of the large side flat surfaces. The experimental research intended to analyze the temperature variation in time during the cooling process, especially in the center of the surface opposite the one being cooled. Numerical modeling used the finite element method with the working conditions mentioned in Table 2. Thermal conductivities values of 0.13 W/(m∙K) for PLA and 0.173 W/(m∙K) for ABS were considered.

Figure 3 shows the temperature distribution of the investigated area after 180 s from the start of the cooling process. The test sample temperature reaches −5 °C in the center and has higher values towards the edges of the test sample. The smaller values of temperature decrease at the edges and corners of the test sample may be explained by the larger influence of the clamping subsystem and the heat gain effect from the outside. 

Eventually, the corners will reach the same temperature as the center area, provided the test sample’s cooling process is long enough. In the case of the ABS material, with a thickness of 3 mm, the temperature difference between the corners and the center area of the test sample is about 1.5 °C. In contrast, for the case of PLA material, the difference is 2.5 °C. The numerical model developed using the finite element method shows that the ABS polymeric material cools faster and is more uniform than the PLA material.

Similar behavior is observed for thinner samples of 1 mm, but the samples cooled faster than the 3 mm thick ones. 

Graphical representations obtained using the finite element method agree with the results obtained from thermal imaging infrared camera (Figure 4). The temperature distribution images from the IR camera provided visual information on temperature variation during the cooling process. The ST 660 Series infrared thermometer performed more accurate temperature measurements at a particular location on the sample surface. Knowing the temperature distribution within the part used for the FEM analysis is important when the parts are subjected to mechanical stress in an environment with large temperature variations.

### 3.2. Experimental Conditions

The main objective of the experimental research was to study the behavior of some 3D-printed test samples of two different polymers subjected to a temperature decrease. Another objective was to model the influence exerted by some factors that characterize the 3D printing process on the temperature decrease rate when one of the test sample surfaces was subjected to a forced decrease in temperature.

An Ultimaker 2 printer was used as 3D printing equipment to manufacture the eight test samples (Figure 5).

The possibility of using an experimental test program was considered [40,41]. A Taguchi L8 program with seven independent variables at two levels of experimentation was preferred, which accepts a monotonous variation of the values of the output parameters of the investigated process, i.e., without the presence of minimums or maximums. The statement may be valid for not-too-large intervals of the input factor values in the investigated process. However, processing the results of the experimental tests by the least-squares method may also lead to the identification of mathematical models for the variation of an output parameter, which is more complex than the mathematical model corresponding to a polynomial of first-degree.

Two levels of variation were adopted for each of the variables considered. In columns 2–8 of Table 2, the coded values of the input factor values were taken into account as independent variables (at the fraction numerator), and the actual values of those factors (at the fraction denominator) were entered as fractions.

The seven independent variables were as follows:The nature of the test sample material. Test samples of polylactic acid (PLA) and acrylonitrile butyl styrene (ABS) with dimensions of 100 mm × 100 mm × (1 or 3) mm were printed, respectively. In Table 2, the values defining the experimental conditions and the values of an output parameter were entered. The PLA material and the ABS material were assigned symbols 1 and symbol 2, respectively. The symbol *m* was used for the material identified as an independent variable;The test sample thickness *h*. The two levels of this factor correspond to a thickness of 1 mm and 3 mm, respectively;Printing speed *s*. The values of this input factor were 45 mm/s and 55 mm/s, respectively. It was considered that the printing speed could affect the arrangement of the molten polymer when the layers of the future test sample were generated, and thus, the thermal conductivity of the deposited material could be affected;The cooling conditions provided by the 3D printer, symbolized by the letter *c* and expressed as a percentage, using the symbol 1 for lack of cooling and 2 for the maximum use of the cooling subsystem of the 3D printer;Level *i* of the infill, for which the values used were 22% (level 1) and 18% (level 2), respectively;The thickness *l* of the layer deposited during 3D printing is 0.06 mm and 0.15 mm, respectively. The values of the input factors that define the working conditions used for the 3D printing process were established by taking into account the recommendations for such a manufacturing process;The size *t* of the time interval at which the temperature measurement was performed.

In Figure 6, the eight test samples (four of PLA and four of ABS) used in the experiment can be observed, with the corresponding data listed in Table 2. After a flexible-wall package containing the gel coolant was placed in contact with one test sample surface, the temperature was measured on the opposite surface at 60 s intervals using an ST 660 Series infrared thermometer.

The IR thermometer can measure temperatures between −50 °C and 999 °C. According to the manufacturer’s specifications, the thermometer has a repeatability of ±1 °C and a resolution of 1 °C.

To verify the accuracy of the IR thermometer, test measurements were performed to determine the temperature at the bottom of an aluminum container with a 30 mm thick layer of boiling distilled water. The temperature indicated by the infrared thermometer from a distance of 250 mm was 99 °C. The experimental results were consistent with the device characteristics provided by the manufacturer, even if possible influences of the presence of water, the variation of the water layer thickness due to the boiling, and the thermal behavior of the aluminum container’s bottom wall are taken into account.

First measurements of temperature variation were taken (as shown in the schematic representations in Figure 1 and Figure 2) to establish the magnitude of the time interval required to determine the empirical mathematical model. As the parameter of interest was the ability of the test sample material to sense the temperature decrease, measurements were performed until, after the initial decrease, the temperature remained somewhat constant for a short time and then began to rise. The results of the measurements are listed in Table 3. 

A graphical representation of the temperature decrease measured on the free surface of the test sample can be seen in Figure 7. The analysis of the data in Table 3 and Figure 7 defined the initial 120 s as the optimum time interval that could provide useful information on the thermal conductivity of the test sample material by a certain temperature decrease *Δ**θ*.

The graph in Figure 7 shows that for the time interval between 0 and 120 s, the temperature of the test sample free surface decreased continuously, even reaching the approximately constant temperature period before starting to grow again. For each test sample made of the two materials (PLA and ABS) and each experiment, the values *Δ**θ* of the temperature decrease were entered for 0–120 s in column 9 of Table 2.

## 4. Discussion

The values of the input factors taken into account and the values *Δ**θ* of the temperature decrease were introduced in a specialized software designed to identify an empirical mathematical model according to the results of the experimental tests [42]. This program is based on the least-squares method. It allows the selection of the most appropriate empirical mathematical model from five such possible models (first-degree polynomial, second-degree polynomial, power type function, logarithmic function, and hyperbolic function). The selection of the most appropriate program is based on the value of the so-called Gauss criterion. 

The value of the Gauss criterion is calculated as a sum of the squares of the ordinate differences corresponding to the experimental results and those determined using the empirical mathematical model considered for the same values of the input factors. The lower the value of Gauss’s criterion, the better the empirical mathematical model considered agrees with the experimental results.

It was found that, among the five versions of empirical mathematical models, the most appropriate concerning the experimental results is the first-degree polynomial type, which has the form:(2)Δθ=39.916−3.999m−5.999h+0.200s−0.00499c+0.124i−94.444l+0.0374t,
where the value of Gauss’s criterion is *S_G_* = 2.379693 × 10^−9^.

Note that this first-degree polynomial provides direct information about the magnitude and direction of the variation of the output parameter (Δ*θ*) to the variation of each input factor value by analyzing the values of the coefficients attached to each of the input factors considered. A similar property is presented by the mathematical model of the power-type function. Power-type empirical mathematical models have often been preferred (for example, to highlight the influence of cutting conditions on tool life, surface roughness, cutting force component sizes, etc.). For this reason, it was considered useful to take into account, for the analyzed situation, an empirical mathematical model of the power function type, which has the form:(3)$$\Delta \theta =0.214m^{-0.336}h^{-0.457}s^{0.691}c^{0.00415}i^{0.512}l^{-0.411}t^{0.016},$$
where the value of the Gauss’s criterion is *S_G_* = 2.430406∙× 10^−7^, a value higher than that when using the mathematical model of the first-degree polynomial type, identified by the software used as the most suitable for the experimental results.

The last two columns (10 and 11) in Table 2 list the values calculated for those two empirical mathematical models. Thus, the extent to which the values determined by using the two types of empirical mathematical models (first-degree polynomial and power-type function) approach the values determined by experimental research may be observed.

Analysis of the differences between the real values of the temperature decrease Δ*θ* and the values determined using the first-degree polynomial empirical mathematical model shows that these errors are between the limits of 0.19 °C (for experiment no. 7) and 2.53 °C (for experiment no. 1). One may observe the relatively low differences between the values obtained experimentally and those determined using the proposed empirical mathematical models, highlighting the empirical models’ adequacy to the experimental results.

Using the mathematical model based on Equation (2), the graphical representations from Figure 8, Figure 9 and Figure 10 were elaborated.

Thus, an increase in the values of printing speed *s*, infill *i*, deposited layer thickness *l*, and time *t* increases the temperature *Δ**θ* because the values of coefficients attached to the respective input factors in the first-degree polynomials are positive. At the same time, an increase in the test sample thickness *h* and the characteristic related to cooling conditions *c* lead to a reduction in the temperature decrease *Δ**θ*, which is highlighted by the negative values of the coefficients associated with the factors concerned in Equation (2). One cannot obtain qualitative information based on the value of factor *m*, as this factor is used only to differentiate the results obtained for the two test samples materials (PLA and ABS).

Among all six factors considered and whose values can be modified, the strongest influence is exerted by the deposited layer thickness *l* and by the test sample thickness *h*, which corresponds, in this order, to the highest absolute values of the coefficients attached in the first-degree polynomial empirical mathematical model (2). It was also expected that an increase in the test sample thickness *h* would reduce the temperature decrease *Δ**θ* for the same time interval due to the larger distance traveled by the heat flow. A minimal influence is exerted by the factor *c*, which considers the cooling conditions. This is emphasized by the low value of the coefficient attached to this factor in Equation (2) and thus has virtually no influence on the output parameter taken into account.

The diagram in Figure 10 depicts the evolution of the temperature decrease *Δ**θ* as a function of time *t* for the two materials, according to the empirical mathematical model constituted by Equation (2). As expected, the increase in time *t* results in an increase in temperature decrease *Δ**θ*, but values recorded for the ABS material are lower than those in the case of the PLA material, which shows a lower thermal conductivity of the latter material.

## 5. Conclusions

The thermal conductivity of materials for parts made by 3D printing is important when used for the thermal insulation of some spaces affected by temperature variations. The research highlighted the transmission of temperature through various 3D-printed plates of polylactic acid (PLA) and acrylonitrile butadiene styrene (ABS), respectively. On one surface of the test sample, a flexible-wall package with a gel coolant at −15 °C was placed. The influence of the thickness of the test samples and the values of some input factors in the 3D printing process on the decrease of the temperature on the opposite surface were taken into account. Equipment to ensure conditions for fixing the test sample inside a box of insulating material, placing a flexible-wall package, and temperature measuring possibilities was designed and fabricated. Following the recommendations for a Taguchi type L8 factorial experiment, experimental tests were performed with 2^7−4^ = 8 experimental tests. The experiments involved using seven input factors at two levels of variation. By mathematical processing of the experimental results, using specialized software based on the least-squares method, an empirical mathematical model of the first-degree polynomial type was identified. The values of the coefficients attached to each of the input factors in this first-degree polynomial provide information on the variation direction, and the magnitude of the temperature decrease when the values of the input factors change within certain limits. The analysis of the empirical mathematical model and graphic representations highlighted that the ABS material ensures better thermal insulation conditions than the PLA material. Among the input factors in the 3D printing process, the strongest influence is exerted by the deposited layer thickness, test sample thickness, and printing speed. The cooling conditions during the 3D printing process have a lesser influence on the decrease in temperature over time.

In the future, it is intended to expand the theoretical and experimental research on thermal conductivity and other materials used in parts manufacturing by 3D printing.

## Figures and Tables

**Figure 1 polymers-14-01714-f001:**
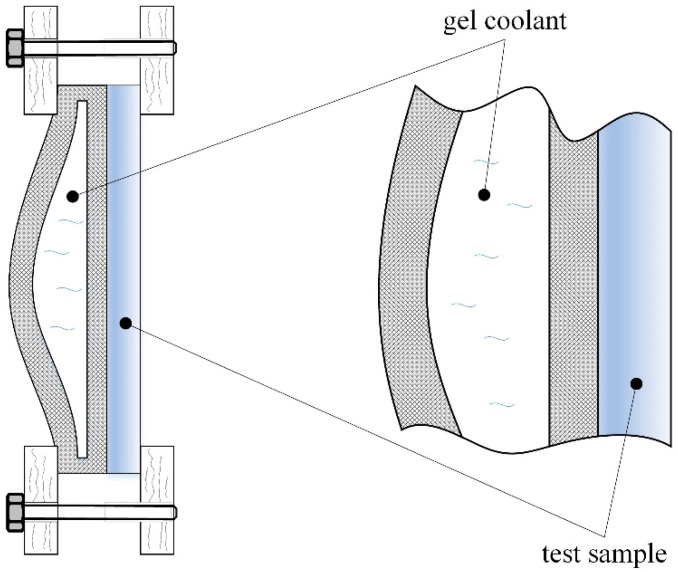
Clamping of the flexible-wall package with gel coolant on the test sample side.

**Figure 2 polymers-14-01714-f002:**
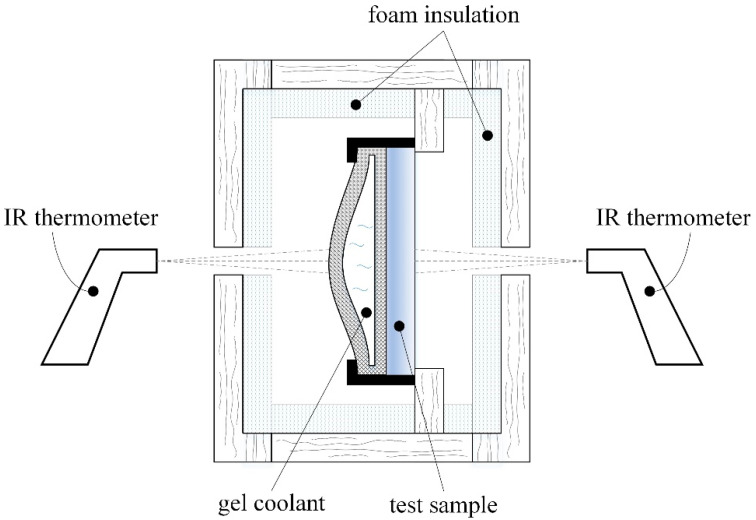
Schematic representation of the experimental arrangement.

**Figure 3 polymers-14-01714-f003:**
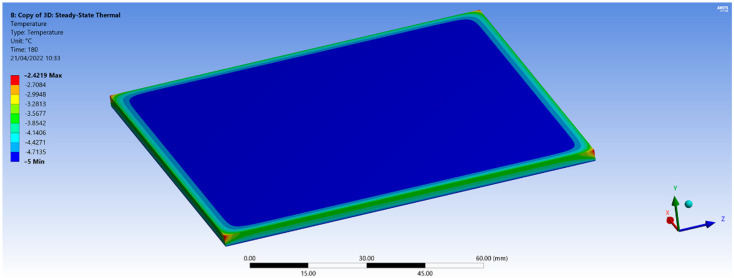
Result of finite element modeling of the temperature variation in the sample test plate (modeling using ANSYS, PLA specimen thickness: 3 mm).

**Figure 4 polymers-14-01714-f004:**
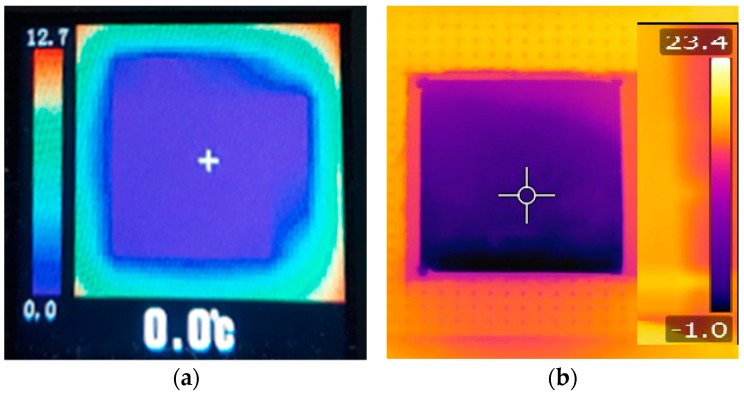
Images of the free surface of the test sample obtained using thermal imaging cameras: (**a**) image obtained using the thermal imaging camera AMG8833, after 180 s from the start of the cooling process for the test sample no. 3; (**b**) image obtained using the infrared camera FLIR T630 SC after 120 s from the start of the cooling process for the test sample no. 8.

**Figure 5 polymers-14-01714-f005:**
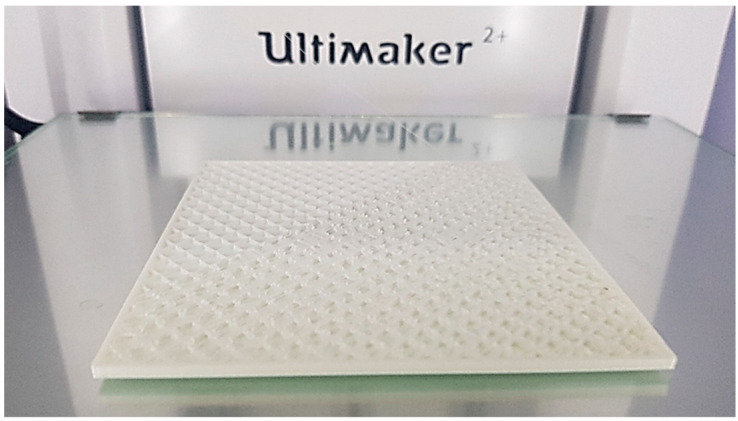
Image while printing a test sample (experiment no. 8).

**Figure 6 polymers-14-01714-f006:**
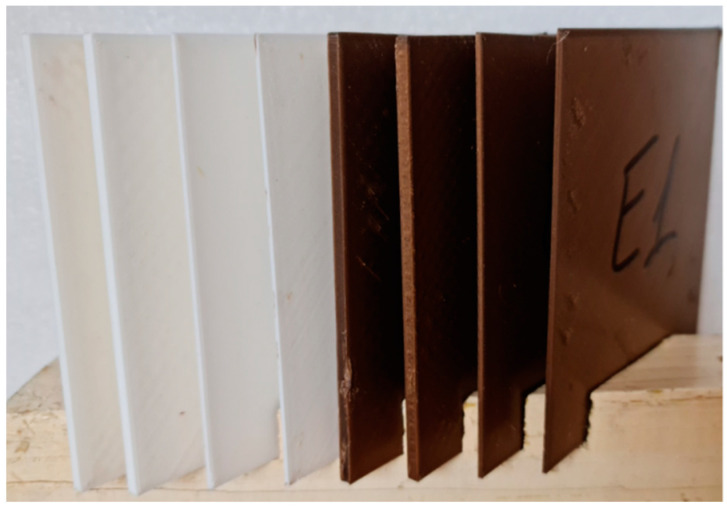
Test samples obtained by 3D printing of ABS (white) and PLA (brown) used to evaluate the temperature decrease in a certain time interval.

**Figure 7 polymers-14-01714-f007:**
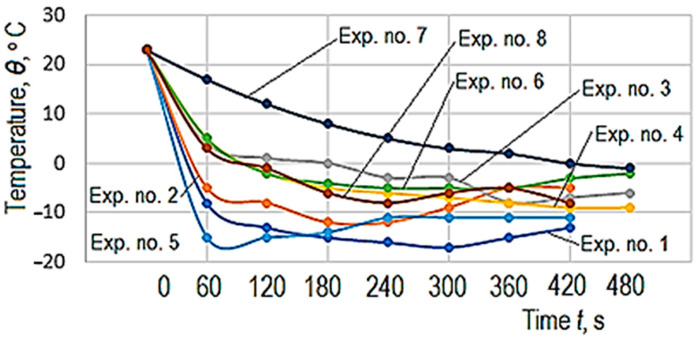
Graphical representation of the temperature decrease on the surface of the test sample opposite to that affected by a cooling process, for all 8 experimental tests, according to the values entered in Table 3.

**Figure 8 polymers-14-01714-f008:**
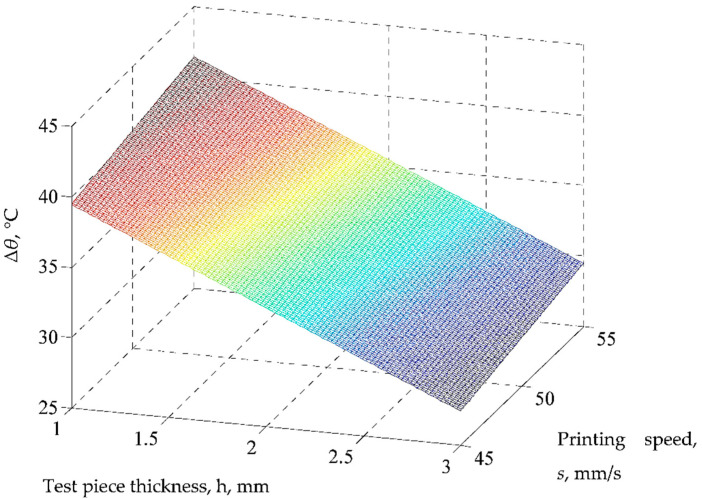
Influence of test sample thickness *h* and printing speed *s* on the temperature decrease *Δ**θ* (*m* = 1 (PLA), *c* = 100%, *i* = 18%, *l* = 0.06 mm, *t* = 120 s; the differences between the real values and those implied by the proposed empirical mathematical model are between the limits of 0.19 °C (for experiment no. 7) and 2.53 °C (for experiment no. 1)).

**Figure 9 polymers-14-01714-f009:**
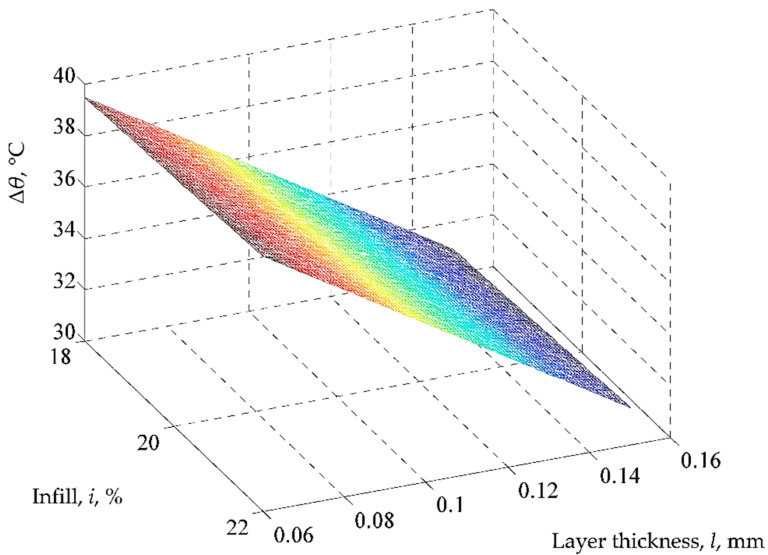
Influence of infill *i* and layer thickness l on temperature decrease *Δ**θ* (*m* = 1 (PLA), *h* = 1 mm, *s* = 45 mm/s, *c* = 100%, *t* = 120 s; the differences between the real values and those implied by the proposed empirical mathematical model are between the limits of 0.19 °C (for experiment no. 7) and 2.53 °C (for experiment no. 1)).

**Figure 10 polymers-14-01714-f010:**
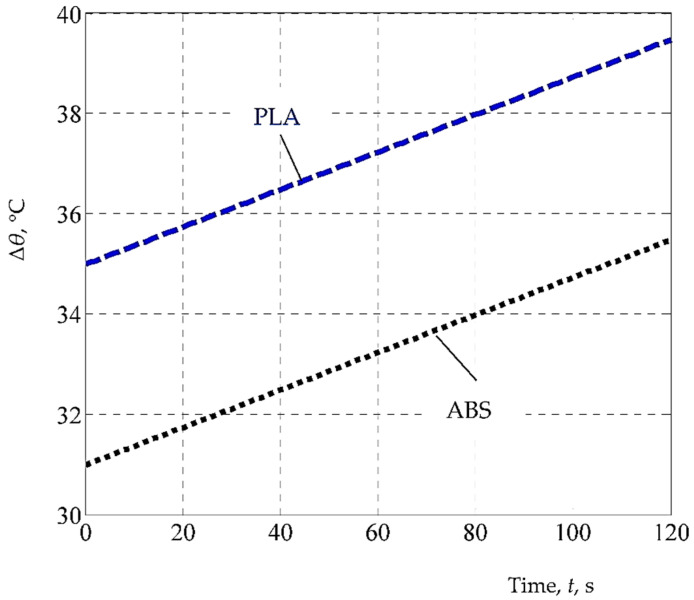
The influence exerted by time *t* on the temperature decrease *Δ**θ* for the two mate-rials considered (*h* = 1 mm, *s* = 45 mm/s, *c* = 100%, *i* = 18%, *l* = 0.06 mm; the differences between the real values and those implied by the proposed empirical mathematical model are between the limits of 0.19 °C (for experiment no. 7) and 2.53 °C (for experiment no. 1)).

**Table 1 polymers-14-01714-t001:** Some physical properties of the materials of test samples made by 3D printing [37,38].

Physical Property	Material	
	PLA	ABS
Thermal conductivity [W/(m·K)]	0.13	0.173
Density [kg/m^3^]	1240	1040
Glass transition [°C]	57 by DSC 63 by DMA	105 by DSC 108 by DMA
Heat deflection temperature [°C]	49	96
Coefficient of thermal expansion [m/m·K]	41 × 10^−6^	72 × 10^−6^
Heat capacity [J/kg·K]	1800	1670

**Table 2 polymers-14-01714-t002:** Experimental conditions and results for a factorial experimental plan L_8_ (2^7−4^ = 8 experimental tests).

Experiment No.	Input Factors (Coded Value/Real Value)							Output Parameter		
	1 Test Sample Material, *m*(PLA/ABS)	2 Test Sample Thickness, *h,* mm	3 Printing Speed, *s,* mm/s	4 Cooling, *c,* %	5 Infill, *i,* %	6 Deposited Layer Thickness, *l,* mm	7 Time, *t*, min	Real Value, *Δ**θ*, after 120 s	According to the Polynomial of First-Order Empirical Model	According to the Power Type Empirical Model
Column No. 1	2	3	4	5	6	7	8	9	10	11
1	1/PLA	1/1	1/45	1/0	1/22	1/0.06	1/0	36	35.98	38.5
2	1/PLA	1/1	1/45	2/100	2/18	2/0.15	2/120	31	30.97	31.3
3	1/PLA	2/3	2/55	1/0	1/22	2/0.15	2/120	22	21.97	22.5
4	1/PLA	2/3	2/55	2/100	2/18	1/0.06	1/0	25	24.99	24.4
5	2/ABS	1/1	2/55	1/0	2/18	1/0.06	2/120	38	37.97	38.8
6	2/ABS	1/1	2/55	2/100	1/22	2/0.15	1/0	25	24.98	24.3
7	2/ABS	2/3	1/45	1/0	2/18	2/0.15	1/0	11	10.99	10.8
8	2/ABS	2/3	1/45	2/100	1/22	1/0.06	2/120	24	23.97	24.2

**Table 3 polymers-14-01714-t003:** The time variation of the free surface temperature after bringing the opposite test sample surface in contact with the gel coolant package.

Exp. No.	Time, *t*, s											
	0	60	120	180	240	300	360	420	480	540	600	660
1	23	−8	−13	−15	−16	−17	−15	−13				
2	23	−5	−8	−12	−12	−9	−5	−5				
3	23	4	1	0	−3	−3	−8	−7	−6	−5		
4	23	5	−2	−5	−6	−7	−8	−9	−9	−8	−8	−8
5	23	−15	−15	−14	−11	−11	−11	−11				
6	23	5	−2	−4	−5	−5	−5	−3	−2			
7	23	17	12	8	5	3	2	0	−1	0	1	
8	23	3	−1	−6	−8	−6	−5	−8				

## Data Availability

Not applicable.

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
