# Peer review of "Theoretical, Numerical and Experimental Assessment of Temperature Response in Polylactic Acid and Acrylonitrile Butadiene Styrene Used in Additive Manufacturing"

_polymers, 2022, doi:10.3390/polym14091714_

Round 1
Reviewer 1 Report
The title of the paper is too general and needs to be rethought and adapted to the scientific content and finding.
Some sentences are not correct/clear. Hereafter some examples:
- In the last section in the introduction part, (…of test samples made of polylactic acid, respectively from acrylonitrile butadiene styrene.) I don’t understand if materials tested are produced with PLA or ABS ? In many place, the The word 'respectively’ is not always used in the right place. Please check.
- In figure 1, (a) and (b) mentioned in the caption are not identified in the figure.
Figure 7 is complex and difficult to read.
- It could be interesting that authors explain how the printing speed can affect the temperature decrease.
Author Response
Authors' responses to the reviewers' comments
The authors of the reviewed paper wish to express their gratitude for the efforts of the reviewers invested in the analysis of the proposed paper and for the useful observations and suggestions to improve the quality of the paper.
All the changes were highlighted in the manuscript of the paper by using the color green.
REVIEWER 1
Reviewer's comment no. 1. The title of the paper is too general and needs to be rethought and adapted to the scientific content and finding.
Authors response to the reviewer's comment. The authors considered that the reviewer was right. The following article title was proposed: "An empirical mathematical model for predicting the heat transmission by polylactic acid and acrylonitrile butadiene styrene in additive manufacturing".
Reviewer's comment no. 2. Some sentences are not correct/clear. Hereafter some examples:
In the last section in the introduction part, (…of test samples made of polylactic acid, respectively from acrylonitrile butadiene styrene.) I don't understand if materials tested are produced with PLA or ABS ?
Authors response to the reviewer's comment. The authors considered that the reviewer was right. The text of the paper has been corrected.
Reviewer's comment no. 3. In many place, the The word 'respectively' is not always used in the right place. Please check.
Authors response to the reviewer's comment. Corrections have been made to the text of the article. The authors considered that the reviewer was right.
Reviewer's comment no. 4. In figure 1, (a) and (b) mentioned in the caption are not identified in the figure.
Authors response to the reviewer's comment. The authors considered that the reviewer was right. A simplified version of the figure was used, so that it was no longer necessary to specify the components a and b.
Reviewer's comment no. 5. Figure 7 is complex and difficult to read.
Authors response to the reviewer's comment. The authors considered that the reviewer was right. The figure caption has been modified to explain the figure's content better.
Reviewer's comment no. 6. It could be interesting that authors explain how the printing speed can affect the temperature decrease.
Authors response to the reviewer's comment. The authors considered that the reviewer was right. A hypothesis in this regard was introduced when the input factors considered were listed.

Reviewer 2 Report
Dear authors,
The presented manuscript aims at measuring the thermal conductivity of polymers and parts produced by FFF manufacturing. The additive manufacturing processing has been widely used although the thermal conductivity of resulting parts has not been widely studied. The study is relevant and interesting as it proposes a sound strategy to measure the thermal properties of materials.
The presented manuscript details the processing of PLA and ABS materials and the measuring their thermal conductivity. The methodology for thermal conductivity evaluation is presented. However, the description of the materials and methods does not provide sufficient details and information and need further revision.
General comments:
The introduction does not provide references or details of thermal conductivity of polymers. It is mentioned that there are polymers with good thermal conductivity but no references, values or strategies are presented in the manuscript. It is recommended to revise the introduction and complete it with details related to materials and current methods to measure the thermal conductivity which are presented.
PLA and ABS are used as materials for the study. However, there is no information regarding the providers of the materials and the quality of the filaments used. It is necessary to identify any additive, plastisizer or relevant components that may affect the final properties of the printed parts. Moreover, it is recommended to change PLA instead of P.L.A. in the whole manuscript.
It is very important to define the equipment used for measuring the temperature by IR and to mention the precision and reproducibility of the measurements as they are critical. The precision of the measurement may be affected by the different behaviour of the materials to the IR. Please provide calibration protocols or discuss this approach. Is it possible to use thermocouples?
In figure 6, some specimens are presented. However, it is not clear which plate correspond to each material. None of the virgin materials (PLA and ABS) correspond to the colors showed in the figure. Please clarify which material and grade have been used. The specimens correspond to some modified grades or materials with some additives. This is very important for the discussion of the results.
It is also important to include the errors in the different graphs to present the results and discuss them. Please comment on the number of replicates performed in each sample and condition.
Finally, variations of the thermal conductivity of the materials are discussed depending on the processing conditions. However, no absolute values are presented in the manuscript. It would be interesting to measure the thermal conductivity of the printed specimens by conventional methods and compare with the obtained results so the manuscript will offer an added value to the potential readers of Polymers.
My recommendation to the editor would be to reject the manuscript in its current form and include all the comments.
Author Response
Authors' responses to the reviewers' comments
The authors of the reviewed paper wish to express their gratitude for the efforts of the reviewers invested in the analysis of the proposed paper and for the useful observations and suggestions to improve the quality of the paper.
All the changes were highlighted in the manuscript of the paper by using the color green.
REVIEWER 2
Reviewer's comment no. 1. The presented manuscript details the processing of PLA and ABS materials and the measuring their thermal conductivity. The methodology for thermal conductivity evaluation is presented. However, the description of the materials and methods does not provide sufficient details and information and need further revision.
Authors response to the reviewer's comment. The authors considered that the reviewer was right.
Additional information has been provided in the article regarding the two materials used in the experimental tests.
Reviewer's comment no. 2. The introduction does not provide references or details of thermal conductivity of polymers. It is mentioned that there are polymers with good thermal conductivity but no references, values or strategies are presented in the manuscript. It is recommended to revise the introduction and complete it with details related to materials and current methods to measure the thermal conductivity which are presented.
Authors response to the reviewer's comment. The authors considered that the reviewer was right. Additional explanations have been introduced in the text of the article in connection with current methods and equipment for measuring thermal conductivity.
Reviewer's comment no. 3. PLA and ABS are used as materials for the study. However, there is no information regarding the providers of the materials and the quality of the filaments used. It is necessary to identify any additive, plastisizer or relevant components that may affect the final properties of the printed parts.
Authors response to the reviewer's comment. The authors considered that the reviewer was right. Additional information has been provided in the article regarding the aspect mentioned by the reviewer.
Reviewer's comment no. 4. Moreover, it is recommended to change PLA instead of PLA in the whole manuscript.
Authors response to the reviewer's comment. The authors considered that the reviewer was right. Only the symbols PLA and ABS were used in the text of the paper.
Reviewer's comment no. 5. It is very important to define the equipment used for measuring the temperature by IR and to mention the precision and reproducibility of the measurements as they are critical. The precision of the measurement may be affected by the different behavior of the materials to the IR.
Authors response to the reviewer's comment. The authors considered that the reviewer was right. Additional explanations on the resolution and repeatability of the measurements made using the infrared thermometer have been mentioned in the article.
Reviewer's comment no. 6. Please provide calibration protocols or discuss this approach.
Authors response to the reviewer's comment. The authors considered that the reviewer was right. Verifying the accuracy of the measurement of the infrared thermometer was possible by measuring the temperature in the bottom of an aluminum container containing a 30 mm thick layer of boiling distilled water. The temperature indicated by the infrared thermometer at a distance of 250 mm from the bottom of the aluminum container was 99 o C. The experimental result thus determined was consistent with the characteristics of the device indicated by the manufacturer, even if the possible influences of the presence of water, the variation of the thickness of the water layer due to the boiling of the liquid, and the thermal behavior of the bottom of the aluminum container are taken into account.
A text with content identical to those mentioned above has been inserted into the manuscript.
Reviewer's comment no. 7. Is it possible to use thermocouples?
Authors response to the reviewer's comment. An instrument that uses thermocouples to evaluate the temperature could be used in the case of experimental tests. Still, we appreciated that the experiment duration would be a little longer, and it was easier for us to use the infrared thermometer available.

Reviewer 3 Report
Kindly consider the attached file

Author Response
Authors' responses to the reviewers' comments
The authors of the reviewed paper wish to express their gratitude for the efforts of the reviewers invested in the analysis of the proposed paper and for the useful observations and suggestions to improve the quality of the paper.
All the changes were highlighted in the manuscript of the paper by using the color green.
REVIEWER 3
Reviewer's comment no. 1. Introduction: The bibliography and included papers is not sufficient and it is too simple. In order to improve the impact of your work, and thus discuss the different related works around your subject, it is necessary to consider the following suggestions: I propose to include some references regarding the application of 3D printing process, as an example in line 57-60, you can explain the application of 3D printing in different fields. There are numbers of papers that you can consider them. Regarding the application of 3D printing in biomedical, please consider the following reference:
https://doi.org/10.3390/polym13244442
Authors response to the reviewer's comment. The authors took into account the reviewer's comments.
Reviewer's comment no. 1 Another remark refers to the fact that although you have worked on the conductivity, there is not any specific explanations on the importance of heat transfer. I propose to add some explanation particularly for the works done on the heat transfer of PLA and ABS. Please consider the following references for heat transfer of PLA:
https://doi.org/10.1016/j.jmapro.2022.02.042
Authors response to the reviewer's comment. The authors took into account the reviewer's comments by introducing additional explanations in the manuscript's text.
Reviewer's comment no. 2. Figure 1: Is there any reference for the figure or it is schematic from authors? It should be clearly mentioned as you are writing a scientific paper.
Authors response to the reviewer's comment. In the publication of scientific articles, there is the convention that when a text, a figure, a table, an equation, or an idea from another work is taken, there should be a reference to the bibliographic source used. Since, in the case of Figure 1, there is no such a reference, this means that the authors of the paper elaborated on the figure.
Reviewer's comment no. 3. Table 1: Have you verified the glass transition temperature that has been determined by DMA, whether it is based on the standards or maybe different frequencies?
Authors response to the reviewer's comment. A glass transition temperature check was not performed, as the information in Table 1 was entered taking into account the information included in the indicated bibliographic source.
Reviewer's comment no. 4. The abbreviations for PLA and ABS are not correct.
Authors response to the reviewer's comment. The reviewer is right. The spelling of the two abbreviations has been changed in the paper.
Reviewer's comment no. 5. Figure 4: Is there any information regarding the calibration of the camera for measuring the temperature ?
Authors response to the reviewer's comment.
The thermal imaging infrared camera was used to obtain, first of all, an image of the temperature distribution on the free surface of the sample test and of the way it evolves during the cooling process. To obtain the empirical model, the more accurate temperature measurement at a particular point on the free sample surface was done with the ST 660 infrared thermometer. For this type of thermometer, a calibration protocol was developed, an aspect that was specified in the new version of the paper.
Reviewer's comment no. 6. Figure 7: I propose to add legends for the curves instead of this type of presenting the different conditions.
Authors response to the reviewer's comment. We considered the reviewer to be right. We have modified the content of the legend in the case of Figure 7.
Reviewer's comment no. 7. Figure 8 and 9: the curves are not clear and I propose to use a colorful curve for better presentation of the results.
Authors response to the reviewer's comment. We considered that the reviewer was right. The mentioned figures have been remade
Reviewer's comment no. 8. In figure 6, some specimens are presented. However, it is not clear which plate correspond to each material. None of the virgin materials (PLA and ABS) correspond to the colors showed in the figure. Please clarify which material and grade have been used. The specimens correspond to some modified grades or materials with some additives. This is very important for the discussion of the results.
Authors response to the reviewer's comment. The authors considered that the reviewer was right. Additional information has been entered in the figure caption.
Reviewer's comment no. 9. It is also important to include the errors in the different graphs to present the results and discuss them.
Authors response to the reviewer's comment. An emphasis on the magnitude of the differences between the real values of temperature decreases and ("Analysis of the differences between the real values of the temperature drop Δθ and the values determined using the first-degree polynomial empirical mathematical model shows that these errors are between the limits of 0.19 o C (for experiment no. 7)" and 2.53 oC (for experiment no. 1)").
In the captions of Figures 8, 9 and 10, the following statement was also introduced: "the differences between the real values and those implied by the proposed empirical mathematical model are between the limits of 0.19 o C (for experiment no. 7) and 2.53 o C (for experiment no. 1)".
Reviewer's comment no. 10. Please comment on the number of replicates performed in each sample and condition.
Authors response to the reviewer's comment. Only one experimental test was used for each combination of input factor values (according to the requirements of a fractional factorial experiment, with 7 independent variables and three levels of variation). If it is considered that the 8 experimental tests are too few, we will show that the method of programming factorial experiments has been proposed precisely to reduce the number of experimental tests that should be performed to identify an empirical mathematical model. In one of the many works that also addresses the issue of factorial experiments (Wayne W. Daniel, Chad L. Cross, Biostatistics: A Foundation for Analysis in the Health Sciences, John Wiley & Sons, 2018, ISBN: 978-1-119-49657- 1), in chapter 8.5 (" The Factorial Experiment"), the authors show that "The advantages of the factorial experiment include the following: 1. The interaction of the factors may be studied. 2. There is a saving of time and effort.".
Reviewer's comment no. 11. Finally, variations of the thermal conductivity of the materials are discussed depending on the processing conditions. However, no absolute values are presented in the manuscript.
Authors response to the reviewer's comment. Considering that the reviewer was right, the article's title has been modified to better correspond to the article's content. The article's content was aimed at highlighting the influence exerted by some factors on the temperature decrease in the case of test samples made of two materials used for the manufacture of parts by a 3D printing process.
In the relatively short time available to make the various corrections in the article, the authors failed to identify a solution to respond adequately to the reviewer's request. The article's authors thank the reviewer for the problem/suggestion made and will try to find a better solution to this problem.
Reviewer's comment no. 12. It would be interesting to measure the thermal conductivity of the printed specimens by conventional methods and compare with the obtained results so the manuscript will offer an added value to the potential readers of Polymers
Authors response to the reviewer's comment. In principle, the reviewer is right. As mentioned in the reviewer's previous comment, the research's main objective was to model the influence of some factors on the temperature decrease in the case of test samples made of two polymers by 3D printing. As mentioned, the title of the article has been changed. We have not been able to identify a solution to respond appropriately to the reviewer's suggestion. However, the issue reported by the reviewer will remain in our attention in the next period.

Round 2
Reviewer 2 Report
Dear authors,
Thanks for considering all the comments from the different reviewers in your manuscript. The document is more focused and better structured. Although there are still some points that need the attention of the authors the manuscript is relevant for potential readers of the journal.
My recommendation to the editor will be to accept the manuscript in its current form.
I encourage the authors to continue their research and to further explore the potential of additive manufacturing.
With kind regards
Reviewer 3 Report
Best wishes in your work